# Reconstitution of T Cell Subsets Following Allogeneic Hematopoietic Cell Transplantation

**DOI:** 10.3390/cancers12071974

**Published:** 2020-07-20

**Authors:** Linde Dekker, Coco de Koning, Caroline Lindemans, Stefan Nierkens

**Affiliations:** 1Princess Máxima Center for Pediatric Oncology, Utrecht University, Heidelberglaan 25, 3584 CS Utrecht, The Netherlands; l.dekker-11@prinsesmaximacentrum.nl (L.D.); C.A.Lindemans@prinsesmaximacentrum.nl (C.L.); 2Center for Translational Immunology, University Medical Center Utrecht, Utrecht University, Heidelberglaan 100, 3584 CX Utrecht, The Netherlands; C.C.H.deKoning@umcutrecht.nl

**Keywords:** allogeneic hematopoietic cell transplantation, hematological malignancies, immune reconstitution, T cell subsets, serotherapy, conditioning, immunosuppressive therapies, biomarkers

## Abstract

Allogeneic (allo) hematopoietic cell transplantation (HCT) is the only curative treatment option for patients suffering from chemotherapy-refractory or relapsed hematological malignancies. The occurrence of morbidity and mortality after allo-HCT is still high. This is partly correlated with the immunological recovery of the T cell subsets, of which the dynamics and relations to complications are still poorly understood. Detailed information on T cell subset recovery is crucial to provide tools for better prediction and modulation of adverse events. Here, we review the current knowledge regarding CD4^+^ and CD8^+^ T cells, γδ T cells, iNKT cells, Treg cells, MAIT cells and naive and memory T cell reconstitution, as well as their relations to outcome, considering different cell sources and immunosuppressive therapies. We conclude that the T cell subsets reconstitute in different ways and are associated with distinct adverse and beneficial events; however, adequate reconstitution of all the subsets is associated with better overall survival. Although the exact mechanisms involved in the reconstitution of each T cell subset and their associations with allo-HCT outcome need to be further elucidated, the data and suggestions presented here point towards the development of individualized approaches to improve their reconstitution. This includes the modulation of immunotherapeutic interventions based on more detailed immune monitoring, aiming to improve overall survival changes.

## 1. Introduction

Allogeneic (allo) hematopoietic cell transplantation (HCT) has evolved into the primary and potentially curative treatment procedure for patients with high-risk hematologic malignancies. Hematopoietic cells can be derived from bone marrow (BM), cord blood (CB) or peripheral blood (PB), from either matched unrelated or related donors. The first successful allo-HCT, treating a pediatric patient with lymphoma, occurred in 1975. Although much improvement has been made since then, current long-term survival rates are still around 50–65% due to relapsed disease and adverse effects associated with the procedure that might lead to severe and life-threatening conditions. Risk factors involve graft rejection, acute and chronic graft-versus-host-disease (GvHD) and viral reactivations (VR) [1,2,3,4]. These complications have been reported to be a consequence of the chemotherapy or transplant preparative regimens, leading to immune dysregulation and to protracted lymphopenia [5,6]. In addition, the use of T cell-depleting (TCD) serotherapy, such as anti-thymocyte globulin (ATG), in order to decrease the probability of GvHD may have a major impact on immune reconstitution (IR) and therefore affects the risk of VR and relapse [7,8]. IR after allo-HCT of myeloid or natural killer (NK) cells is more rapid compared to the slow reconstitution of T cell populations [2,9,10,11]. Furthermore, T cells often show a skewed T cell receptor (TCR) repertoire and remain dysfunctional even after the recovery to normal lymphocyte numbers [12]. Recent studies provide evidence that T cell reconstitution is key in the development of transplantation-related complications and the patient’s ability to defeat these complications [4,13,14,15,16]. Therefore, an understanding of the processes involved in T cell reconstitution is critical for protection against opportunistic infections, a sustained graft-versus-leukemia (GvL) effect, and survival chances after allo-HCT [2,3,4,13]. Here, we will review the current understanding of T cell reconstitution following allo-HCT as treatment for hematological malignancies. We discuss the post-HCT dynamics of different T cell subsets: CD4^+^ and CD8^+^ αβ T cells, γδ T cells, iNKT cells, regulatory T cells (Tregs), MAIT cells and the reconstitution of both naive and memory cells.

## 2. T Cell Reconstitution after allo-HCT

IR of the T cell compartment after HCT is complex and dynamic. T cell reconstitution involves two phases: homeostatic peripheral expansion (HPE) and thymopoiesis. Initial lymphoid immunity is provided by passenger mature naive and memory T cells that immediately undergo HPE to replenish the T cell compartment. HPE is influenced by either positive or negative T cell selections, cell source, cytokine exposure and TCR stimulation [12,17,18]. This thymus-independent mechanism is mainly important for early T cell reconstitution, since thymopoiesis takes at least 6 to 12 months to occur. Thymopoiesis is affected by age-related regeneration capacity, therapy-induced cytotoxic insults, stem cell source and GvHD [19,20,21,22]. This process results in the emergence of novel phenotypically naive T cells that have maturated in the thymus, simultaneously increasing TCR diversity, which is related to a better clinical outcome [23,24,25,26].

T cell subsets reconstitute in distinct ways post-HCT (Figure 1), which is heavily influenced by multiple transplantation and patient-related factors, including the conditioning regimen [12], cell source [27,28,29], donor type [30], age of recipient and donor [12], HLA mismatches [31,32], infections [33], graft manipulation [17,34], as well as GvHD type, treatment and prophylaxis [9]. T cell reconstitution can therefore even be delayed for over 2 years [9,23,35], which is highly related to morbidity and mortality [2,3,4,13,14,15,16,36,37].

## 3. CD4^+^ T Cells

Naive CD4+ T cells can differentiate into particular lineages based on cytokine stimulation, cytokine milieu, co-stimulation and antigen concentration [38]. The CD4+ T cell compartment is commonly divided into regulatory T (Treg) cells and conventional T helper (Th) cells. Higher numbers of CD4+ T cells after transplantation attenuate GvHD, prevent VR and VR-associated mortality and are significantly correlated with increased relapse-free and overall survival [3,13,33,39,40]. Both subsets react differently to homeostatic signals and therefore reconstitute in distinct ways after allo-HCT.

### 3.1. Th Cells

In many studies, T-helper cells are referred to as total CD4^+^ T cells. After allo-HCT, a very quick but stable recovery of CD4^+^ T cells over time is associated with low incidence of viral reactivations and relapse [39,41,42,43], and with increased overall survival [2,3,4,44]. However, a peak of CD4^+^ T cell levels during the first 90 days after BMT and CBT is correlated to higher mortality [2,45], which is probably due to an underlying event that could potentially affect long term outcome, such as a GvHD or VR.

There are major differences in CD4^+^ T cell reconstitution between recipients receiving grafts from different cell sources [2,3]. CD4^+^ T cells recover within as early as 1–2 months and reach reference levels about 7–12 months after both CBT and BMT, with a better reconstitution after CBT [17,22,41,42,46,47]. CD4^+^ T cells reconstitute more rapidly in patients receiving PB grafts compared to BM grafts [3,4,45,48]. In contrast, it takes up to two years for the CD4^+^ T cells to reconstitute in patients receiving TCD grafts [17,25,47], which contain on average >100-fold lower CD3^+^ cells compared with CB grafts [17]. This is consistent with the principle of HPE as a mechanism driving early T cell expansion, while a better thymic-dependent mediated naive T cell recovery is important for late T cell reconstitution.

Early CD4^+^ T cell reconstitution is highly affected by components of the conditioning regimen. Their reconstitution is extremely delayed when TCD, such as ATG, is used in the conditioning regimen [25,41,44,46,49,50,51]. ATG is included to decrease the chance of developing GvHD, but overexposure may result in severely delayed IR. The results from a recent randomized trial showed that individualized ATG dosing based on weight and total lymphocyte count (TLC) enhanced the CD4^+^ T cell reconstitution post-HCT, as well as overall survival (Trial NL4836, unpublished). The effect of ATG is dramatically influenced by the order of other components in the regimen, as shown in patients receiving total body irradiation (TBI) and cyclophosphamide (Cy) [50]. TBI followed by Cy results in a much more reduced TLC, causing ATG overexposure post-HCT and thus slower IR. Cy followed by TBI on the other hand did not result in ATG overexposure, resulting in a better outcome. Furthermore, Filgrastim (G-CSF), which is routinely used after CBT and not after BMT, dramatically enhances killing of ATG-coated cells [42]. The effect of ATG was first thought to be particularly on naive CD4^+^ T cells [6,44,52], which are present in a larger number in CB grafts [29]. Patients receiving CB grafts without ATG conditioning show a rapid CD4^+^ T cell reconstitution (faster than BM) [40,41,49], which can be associated with the finding that fetal naive CD4^+^ T cells proliferate much more upon stimulation compared to adult naive CD4^+^ T cells [29]. Notably, G-CSF without ATG might positively influence CD4^+^ T cell reconstitution by increasing innate IR, as a strong association between an enhanced innate recovery and CD4^+^ T cell reconstitution after BMT and CBT is described [53]. Furthermore, G-CSF-mobilized grafts have a faster CD4^+^ T cell reconstitution compared to immobilized grafts [45,54]. However, to what extent G-CSF treatment would affect reconstitution of CD4^+^ T cells is unclear. Proliferation of innate immune cells is already higher within CB grafts and innate IR has been shown to be more rapid after CBT compared to BMT/PBT [2,17,53]. In contrast to ATG, posttransplant Cy is suggested to induce T helper cell dysfunction rather than elimination and thymic clonal deletion, thereby reducing both acute and chronic GvHD occurrence [55,56]. Together, adapting the dosing schemes of drugs used as standard-of-care or simply adjusting the order of different treatment modalities can be considered “low hanging fruit” for the improvement of survival chances post-HCT.

In conclusion, CD4^+^ T cell reconstitution can be predicted by various covariates, such as the overlap in timing of residual ATG exposure, innate immune recovery and Filgrastim administration. Hoare et al. (2017) published a mechanistic mathematical model, including many aspects playing a role in IR to predict CD4^+^ T cell reconstitution post-HCT [46]. However, dose and timing of ATG, together with different CD4^+^ T cell subsets, are not included. As the CD4^+^ T cell subsets are responsible for different types of immune responses, a better understanding of the underlying biological mechanisms needs to be obtained using more detailed immune monitoring protocols. Furthermore, prospective cohort studies are needed to increase T cell IR by studying the benefits of personalized conditioning strategies and post-HCT treatment. It is of note that, in most diagnostic labs and clinical studies, CD4^+^ T cells not only include T helper cells but also Tregs. In particular, in the transplantation setting, discrimination of these subsets might be valuable.

### 3.2. Tregs

Tregs (CD4^+^ CD25^+^ FoxP3^+^) comprise 4–10% of the circulating CD4^+^ T cells and maintain immune homeostasis and self-tolerance by inhibiting cytokine secretion and proliferation of antigen processing cells (APCs), NK, B and T cells. They are critical in controlling responses from other immune cell subsets to self and foreign antigens, and play a central role in preventing autoimmune disease. Tregs can be subdivided into naturally occurring Tregs, derived from the thymus, and induced Tregs, which are differentiated from nonregulatory CD4^+^ CD25^+^ cells [1,57]. Induced Tregs are more susceptible to apoptosis and have a less stable expression of FoxP3 [57]. Although debate exists [55,58,59,60], high numbers of FoxP3^+^ T cells in the graft and early post-HCT are negatively correlated with GvHD [18,54,60,61,62,63,64]. However, the literature also suggests that an enhanced Treg function can suppress the GvL effects and thereby the allo-HCT outcome [58].

Reconstitution of Tregs has been suggested to be primarily achieved by HPE without major contribution of thymopoiesis, especially compared to reconstitution of effector T cells [18,51,62,65]. The proportion of Tregs among the CD4^+^ T cell population returns to normal within 6 weeks post-HCT, as soon as the CD4^+^ T cells are detectable [57,60]. In the subsequent 2 to 3 months, differences in the stability of this proportion have been observed [57,60], which might be related to a higher proportion of activation-induced Tregs, the occurrence of GvHD (and associated treatment) and thymopoiesis. This can further be explained by differences in prophylaxis, since sirolimus-based prophylaxis promotes Treg expansion [66] and steroid-treatment as prophylaxis likely enhances Treg prevalence and activity [67]. ATG as GvHD prophylaxis greatly delays reconstitution of Tregs [41,44,49,51], while posttransplant Cy seems not to negatively affect their recovery but rather increases the Treg number and function in murine models, greatly reducing GvHD incidence [55,56,68]. Notably, the exact mechanisms of Cy prophylaxis in humans still need to be elucidated. Despite differences in stability after one-month, normal levels of Tregs are achieved by 9 months post-HCT [18,57,60]. In patients with prolonged CD4^+^ lymphopenia, however, Treg levels were observed to decline after 9 months and remain at very low levels from 12–24 months [18,62]. This might be a result of exhaustion of the Treg pool due to HPE. Together with the reported increase in recent thymic emigrants (RTEs) within the Th population and not within the Treg population upon recovery of thymic function [65], this suggests that effector CD4^+^ T cell recovery plays a role in Treg homeostasis.

Tregs show a more activated phenotype and their reconstitution is more rapid after CBT compared with BMT/PBCT [57], which might be associated with the observation that fetal naive CD4^+^ T cells are more likely to develop into Tregs, unlike adult naive CD4^+^ T cells [29]. Furthermore, steroid-treatment as prophylaxis in CBT might tip the balance to more regulatory responses in patients receiving CB grafts [67]. Together with the lower effector T cell number in CB, this supports the correlation of high Treg numbers with a low GvHD incidence in CBT [69]. Nevertheless, others show no differences in Treg reconstitution between CBT and BMT [60]. Early infusion of donor Tregs without GvHD prophylaxis prevented GvHD [30,70], while others showed correlations between high Treg numbers and GvHD [58,59]. These discrepancies might be explained by the existence of diverse Treg populations and the many parameters affecting associations as covariates, such as conditioning, thymopoiesis, age, adverse events, etc. [18,30,65,71]. In addition, one must realize that the Treg function in different stages post-HCT may differ.

Taken together, these studies suggest a link between the number of Tregs and the development of GvHD. In addition, the exact mechanisms involved in Treg reconstitution, such as the influence of prophylaxis, and their associations with allo-HCT outcome need to be further elucidated. Therefore, studies including well-defined cohorts with Treg subset identification by flow cytometry using staining for CD4^+^ CD25^hi^ CD127^lo^ FoxP3^+^ and not only CD4^+^ CD25^hi^ FoxP3^+^ or CD4^+^ CD25^hi^ together with suppression assays to test the functionality of the Treg cell population are necessary.

## 4. CD8^+^ T Cells

CD8^+^ T cells, often referred to as cytotoxic T cells, provide clearance of virally infected cells and tumor cells by killing them through the release of cytotoxic molecules and cytokines [72]. CD8^+^ T cells are HLA class I-restricted for recognition of antigens and high levels of CD8^+^ T cell counts post-HCT are associated with the chance to develop GvHD [16,45,64,73,74]. However, the alloreactivity by CD8^+^ T cells is also thought to mediate the GvL effect, in particular because a lower CD8^+^ T cell count is associated with higher relapse rates [10,33]. Moreover, higher numbers of CD8^+^ T cells early after transplantation are correlated with increased overall survival [10,31,33,45]. This again illustrates the delicate balance of productive immune function post-HCT to prevent infections and relapse while maintaining functional immune homeostasis and regulation to prevent GvHD.

CD8^+^ T cells reconstitute faster compared to CD4^+^ T cells [7,9,47,65,75], which is influenced by cell source, graft type and cytomegalovirus (CMV) seropositivity [9,31,60,76]. Less studies have focused on CD8^+^ T cell reconstitution compared to reconstitution of CD4^+^ T cells. This might be because the CD8^+^ T cell levels are more variable due to instant reactions to microbial events, decreasing the possibility to find significant correlations [39]. Although thymopoiesis substantially contributes to CD8^+^ T cell reconstitution [5,7], the fast CD8+ reconstitution is presumably a result of the rapid HPE of effector memory CD8^+^ T cells early post-HCT [33,77], as the infused CD3^+^ count and HLA-match are significantly associated with CD3^+^ CD8^+^ T cell recovery [31]. Furthermore, the presence of CMV-specific CD8^+^ effector memory T cells in CMV-seropositive recipients is associated with faster CD3^+^ CD8^+^ T cell recovery, with a markedly faster reconstitution in recipients of grafts from CMV-seropositive donors [76,77,78]. Data on CD8^+^ T cells show that recovery can be as early as 1 month and reconstitution to normal levels within 10 months after transplantation, regardless of cell source and potential damage to the thymus [7,9,33,47,65,75]. When comparing different cell sources, CD8^+^ recovery is faster after PBT [45] than BMT and CBT, in which reconstitution of this compartment is similar [22,60]. Others observed a slow reconstitution following CBT compared with BMT and PBT [2,5,9,17,79], where the proportion reached normal levels 1 year post-HCT. This is comparable to reconstitution in recipients receiving TCD-PB grafts [17,25,47]. However, this delay might be due to the effect of ATG together with G-CSF in CBT [42], although the effect of ATG on CD8^+^ T cells is less compared to CD4^+^ T cells [7,41]. The fast reconstitution among recipients of G-CSF-mobilized PB grafts compared with BM recipients [45] suggests that an increased innate IR due to G-CSF treatment might also have a positive effect on CD8^+^ T cell reconstitution, similar to the effect on CD4^+^ T cell reconstitution [53]. The effect of posttransplant Cy as prophylaxis on CD8^+^ T cells is not yet clear, although this might not directly influence CD8^+^ T cell recovery but is suggested to induce CD8^+^ effector cell dysfunction [55,56].

Overall, CD8^+^ T cell recovery seems to occur more rapid compared to CD4^+^ T cell reconstitution and is highly influenced by cell source and graft type. The fact that CD8^+^ T cell reconstitution correlates with protection against leukemic relapse and infections, and with improved overall survival, underscores the importance of adequate reconstitution of this subset. Further knowledge regarding CD8^+^ T cell recovery and activation obtained through more in-depth immune monitoring might contribute to CD8^+^ T cell reconstitution as a significant predictive variable to timely identify adverse events and the need for immunotherapeutic intervention.

## 5. γδ T Cells

Gamma delta (γδ) T cells normally comprise about 5% of the entire CD3^+^ T cell population and express the γδ TCR instead of the conventional αβ TCR [1,80]. Different from αβ T cells, γδ T cells can rapidly be activated as a response to stress-induced self-ligands that are upregulated on transformed, infected, or otherwise stressed cells [81]. This population is suggested to facilitate allo-engraftment and to exhibit strong anti-infectious and antileukemia-effects, without causing GvHD [33,34,82,83,84]. Although some find correlations with GvHD [33,58,82], high levels post-HCT are correlated with increased leukemia-free survival and overall survival [33,83,84,85]. Due to these properties, both the use of γδ T cells as immunotherapy and post-HCT γδ T cell reconstitution are being more and more recognized within this research area.

Reconstitution of γδ T cells is faster than αβ T cell reconstitution and takes 1–2 months [86,87]. They immediately expand after BMT and haplo-HCT [33,34], which is inversely associated with CD3^+^ and αβ T cell numbers transferred with the graft [84]. Together with an extremely slow reconstitution in patients receiving grafts depleted of both αβ and γδ TCRs [47,83,85], this corroborates the principle of HPE as the mechanism driving early γδ T cell expansion [84,86,87]. Although the γδ T cell population is thought to be mainly derived from HPE throughout the first year post-HCT [33], naive γδ T cells increase in recipients following haplo-HCT between 1 and 3 months [34,87]. This suggests that these cells differentiate from donor hematopoietic stem cells [34,87]. Furthermore, TCR repertoires of regenerated γδ T cells display very different clonotypes from the hosts’ repertoire post-transplantation, supporting de novo development from donor stem cells in the thymus [33]. Although showing a skewed γδ TCR repertoire, this newly established repertoire remains very stable for at least 6 months post-HCT [87]. In addition, no long-term qualitative differences in γδ T cells have been observed between different cell sources [83,85]. Notably, intentionally retaining γδ T cells in the graft, thereby achieving an efficient and fast γδ T cell reconstitution post-HCT, might have a positive effect on HCT outcome.

γδ T cell reconstitution is largely influenced by infections and reactivations, in particular reactivation of CMV [33]. CMV reactivation post-HCT results in rapid and large HPE of specifically CMV reactive γδ T cells [33,34,80,87]. Paradoxically, CMV reactivation is in some studies associated with a reduced risk of relapse [88], possibly because γδ T cells are both capable of recognizing CMV-infected cells and tumor cells of hematopoietic origin. Intentionally reactivating CMV-reactive γδ T cells might therefore have a favorable effect on leukemia relapse risk [34,80]. γδ T cell recovery is further thought to be influenced by immunosuppressive therapies—similar to αβ T cells—although no associations have been found so far [84].

In conclusion, recovery of the γδ T cell compartment is highly influenced by graft type, infections and reactivations. Unfortunately, there are only a limited number of studies that focused on the reconstitution of this important T cell population, which is positively associated with various HCT outcomes. Further prospective cohort studies including a larger number of patients to investigate the driving mechanisms of HPE, the development of newly derived T cells during thymopoiesis, their functionality as well as the effect of the cell source and conditioning regimens on γδ T cell reconstitution and function are needed.

## 6. MAIT Cells

Mucosal-associated invariant T (MAIT) cells comprise 1–10% of CD3^+^ cells and are abundant in both PB and mucosal tissues. They are innate-like T cells that highly express CD161 and a semi-invariant αβ TCRs that recognize microbial metabolites presented by MR1 [89]. MAIT cells can be activated in MR1-dependent and MR1-independent ways, although expansion requires circulating B cells and commensal microbiota [89,90,91,92]. In the allo-HCT setting, no associations between high MAIT cell numbers and protection from infections have been reported so far [93]. However, low frequencies of MAIT cells in the graft and post-HCT are associated with severe GvHD [74,92,93,94,95], and higher numbers seem to be associated with improved overall survival [94].

Reconstitution of MAIT cells post-HCT significantly correlates with age [92,94] and cell source [92,93,96,97]. Early MAIT cell reconstitution is driven by the HPE of the MAIT cells transferred with the graft [93]. Since CB contains much lower frequencies of MAIT cells compared with adult graft sources [97], their reconstitution is highly impaired following CBT [93,96]. Extremely low counts remain up to 12 months [92,96], and normal values after CBT are only reached around 5 years in children [96] and around 10 years in adults [92]. On the contrary, a rapid recovery to a plateau can be seen from Day 30 to Day 100 post-BMT/PBT [93]. MAIT cell frequencies, however, seems to remain much lower compared to healthy controls up to 1–2 years post-HCT [93,98]. Notably, MAIT cells were only monitored within PB and not within mucosal tissues. Furthermore, MAIT cells in CB proliferate upon CD3 stimulation alone, while MAIT cells in adult PB need both CD3 and co-stimulation to proliferate [93,97]. Together with the suggested contribution of thymopoiesis to MAIT cell reconstitution post-CBT [96], this highlights the need for longitudinal studies monitoring MAIT cell frequency and function in both PB and mucosal tissues along with the contribution of thymopoiesis.

MAIT cell reconstitution is positively and negatively influenced by gut microbiota and immunosuppressive therapies, respectively [92,93,98]. The abundance of *Bifidobacterium longum* and *Blautia* spp. post-PBT [93] and gut microbiota diversity post-CBT [92] are positively correlated with a better circulating MAIT cell reconstitution. Their reconstitution seems to be negatively influenced by ATG, cyclosporine A and sirolimus after BMT/PBT [98], and cyclophosphamide after HCT [93]. No associations with incorporation of TBI [93], glucocorticoids and calcineurin inhibitors [92] with MAIT cell recovery were found. However, immunosuppressive therapy-induced proinflammatory signals [89,91], along with an altered gut microbiota composition as a result of conditioning therapy, as well as altered dietary intake and antibiotic use [99], might further influence MAIT cell reconstitution and function after allo-HCT.

Together, MAIT cell reconstitution seems to be extremely slow and depends on age, cell source, gut microbiota and immunosuppression. However, low MAIT cell counts might reflect a migration towards sites of (GvHD-induced) inflammation, although this has not yet been found clinically in humans [96]. Since MAIT cell reconstitution has gained attention in recent years, only a few small studies have focused on circulating MAIT cell recovery. Reproducible methods to detect and quantify MAIT cells and functionally distinct MAIT cell subsets [100] in mucosal tissues are crucially needed to determine the migration of MAIT cells into inflamed tissues.

## 7. iNKT Cells

Invariant NKT (iNKT) cells are rare innate-like T cells with immunomodulatory functions, which express semi-invariant αβ TCRs that recognize lipid antigens presented by CD1d molecules. Similar to MAIT cells, they are capable of secreting large amounts of cytokines upon activation in TCR-dependent and TCR-independent manners [89]. High numbers of iNKT cells in the graft and early after allo-HCT are associated with protection against GvHD [101,102,103,104,105,106] and relapse [103,106,107] and seems to be correlated with improved overall survival [103]. Therefore, using this T cell population as immunotherapy and increasing the iNKT cell numbers after allo-HCT has gained attention in recent years.

Reconstitution of iNKT cells occurs independently of T cells [103] and the proportion of iNKT cells already reaches normal values within 1 month post-HCT [60,108,109]. When comparing distinct cell sources, PB grafts contain higher numbers of iNKT cells compared with BM grafts, and iNKT cells reconstitute faster after PBT compared to BMT [101,105]. Recipients from CB grafts show a slower recovery compared with BM- and PB-transplanted recipients [60,94]. After TCD-HCT, iNKT cells emerged in as early as 3 months, reaching normal reference values by 18 months [107]. Besides the relatively small influence of cell source on iNKT reconstitution, the use of immunosuppressive drugs might have an impact. However, following BMT/PBT, steroid administration seems not to suppress the number of iNKT cells [101] and ATG seems not to impair iNKT cell recovery [109]. Although further studies should investigate whether the slower reconstitution post-CBT is a result of immunosuppressive treatment, iNKT cells seem to display rapid effector functions within 3–6 months post-CBT [108]. This suggests that immunosuppressive drugs might only transiently affect iNKT reconstitution and function immediately after allo-HCT, if at all.

In conclusion, iNKT cells reconstitute early and rapidly following allo-HCT, which is slightly influenced by cell source. The association of iNKT with prevention from GvHD points towards novel therapeutic options to predict or prevent GvHD post-HCT; for example, the adoptive transfer of iNKT cells into recipients that fail to reconstitute this population, or the use of early iNKT/T cell ratios as a new parameter to adapt GvHD prophylaxis. Importantly, iNKT cells comprise distinct subsets with different dynamics following allo-HCT [105,107]. Future studies should focus on these subset dynamics, their relations to clinical outcomes and predictive values for adapting GvHD treatment post-HCT.

## 8. Naive T Cells

The naive T cell (T_n_) compartment (CD45RA^+^ CD45RO^−^ CCR7^+^) consists of a large number of cells with unique TCRs, which potentially proliferate and differentiate into all types of effector and memory progenies upon interacting with newly encountered antigens [110]. This compartment comprises a heterogenous population, including RTEs and mature T_n_ cells. T cell receptor excision circles (TRECs), which decline upon cell division, together with CD31 expression are commonly used to measure RTEs [110,111]. In the allo-HCT setting, early activation of donor T_n_ cells is correlated with chronic GvHD, suggested to be a result of large numbers of alloreactive precursors due to the enormous diversity of the naive TCR repertoire [45,64,65,112,113,114,115]. Overall, adequate T_n_ cell reconstitution is crucial for long term immune function and tolerance [25,116] and correlates to improved overall survival [2,15,45,116].

Early reconstitution of the T_n_ cell pool highly relies on the number of T_n_ cells transferred with the graft [14,22,29,116]. Early after transplantation, T_n_ cells are maintained at relatively normal proportions due to HPE and increased survival [10,65,117], probably resulting in a rapid decline in RTE numbers and reduced TCR diversity [111,118]. Since T_n_ cells are present in a larger number in CB grafts [29], an increase in the percentages of T_n_ cells seems to occur faster early after CBT compared to BMT/PBT [14,21,22,23,111]. Notably, recipients of G-CSF-mobilized PB grafts show higher T_n_ cell numbers compared with BM-transplanted recipients [45]. The fast reconstitution following CBT is probably due to the highly proliferating fetal naive CD4^+^ T cells present in CB grafts [29]. These are poised to become Tregs, decreasing GvHD probability, and seem to mediate a stronger anti-leukemic effect compared to adult T cells [26,29]. In haplo-HCT, however, T_n_ cell activation and numbers might be affected by donor NK cell alloreactivity triggered by HLA mismatches [119,120]. NK cells can become alloreactive when they do not express a certain degree of inhibitory and activating ligands that recognize the HLA class-I alleles on their target cells [32]. NK sensitivity to class-I polymorphism seems to be restricted to hematopoietic cells, thereby impacting T_n_ cell reconstitution due to cytolytic activity against DCs and T_n_ cells, decreasing GvHD occurrence [119,120]. The impact on T_n_ cell reconstitution might persist longer after transplantation, since donor alloreactive NK cells have been detected years after haplo-HCT [120]. Nevertheless, NK cells can also secrete a number of cytokines, which might promote T_n_ cell reconstitution, thereby contributing to GvHD development [32]. Importantly, T_n_ cells transferred with the graft survive posttransplant cyclophosphamide [117,121], while low-dose ATG results in significantly smaller T_n_ cell numbers post-HCT [113,114]. Around 100 days post-HCT, the T_n_ cell pool slowly reconstitute within months to years, mainly accomplished by thymopoiesis [21,22,65,111,118,122].

Thymic differentiation of donor-derived lymphoid progenitors [22,29,122] are thought to be strongly correlated with TREC values [21,26,64,111,116,118,123]. This indicates that the TREC levels reflect real thymic de novo production. Thymopoiesis is significantly influenced by extensive GvHD, immunosuppressive drugs, VR and age [21,22,64,111,115,116,122,124]. Thymic output might be increased as a result of lymphopenia [122]; however, this can also be related to clinical events [115]. When comparing cell sources, TRECs seem to increase faster following CBT compared to BMT/PBT [14,21,23,111]. The presence of a broader TCR repertoire diversity in recipients of CB grafts up to 3 years post-HCT indicates a better thymic reconstitution from CB progenitors [23,24]. Clinically, less TCR diversity post-HCT might indicate the existence of large, dominant donor-derived alloreactive clonotypes, correlating with the increased occurrence of GvHD after BMT and PBT [16]. TREC levels increase similarly following BMT and PBT [116], although recipients of G-CSF-mobilized PB grafts show a higher TREC content compared with BM-transplanted recipients [45]. Depletion of all T cells or only T_n_ cells within PB grafts might not influence thymopoiesis, since RTEs and TCR diversity seems not to be reduced later after transplantation [111,112,118,125]. Notably, others reported a recovery of TRECs to control values already within 6 months post-PBT [115], at which the T_n_ cell numbers are still extremely low [21,112,115]. TREC numbers did not increase after 6 months [26,115], suggesting ongoing cell division [115]. It is therefore essential analyze both TREC content and naive T cell numbers to get insight into the kinetics and dynamics of T cell recovery post-HCT.

Immune reconstitution and restoration of the TCR repertoire later after transplantation requires intact thymic function and usually takes years. Changes in the early T_n_ cell compartment, in particular CD4^+^ naive T cells, presumably predict relapse and other long-term outcomes after allo-HCT. Current RTE markers are useful; however, their association differs between conditions and between CD4^+^ and CD8^+^ T_n_ cells [110]. Furthermore, T_n_ cell reconstitution show distinct dynamics between the T cell subsets [22,64,65]. Longitudinal studies combining immunophenotyping, TRECs measurements and TCR sequencing, together with functional assays, are necessary to provide further insights into the role of the functional heterogeneity of T_n_ cells and their reconstitution in the allo-HCT setting.

## 9. Memory T Cells

Memory T cell subsets show notable plasticity but are generally divided into effector memory (T_EM_), central memory (T_CM_) and stem memory (T_SCM_) T cells [121]. T_EM_ cells mediate stronger effector functions compared to T_CM_ cells [126]; however, T_CM_ cells are much stronger correlated to long term persistence [127,128]. T_SCM_ comprise a naive-like, antigen-experienced, self-renewing population, which is suggested to be positioned upstream from the memory and effector T cell subsets in T cell ontogeny [35,126]. They are able to survive for decades [126,128,129], have enhanced proliferative potential and immune reconstitution capacity, and are therefore thought to be the key source for immunologic memory [126]. The memory T cell compartment is maintained by division of T_SCM_ and through maturation from naive T cells [117,121,129,130,131]. This crucial compartment is depleted in HCT patients, leading to an urgent re-education of immunological memory from the time of Transplant.

Immunological memory transferred with the graft depends on the graft type (with CB containing almost no memory cells), and reconstitution of the memory T cell compartment highly relies on the quality and number of infused memory T cells within the graft [10,35,118]. T_CM_ cells are hardly detectable early after transplantation, probably because they are more sensitive to TCD therapies than T_EM_ and T_SCM_ cells [52]. T_EM_ infused with the graft proliferate and produce cytokines rapidly upon stimulation, and high levels are therefore correlated to a lower incidence of opportunistic infections and a higher incidence of both GvHD and GvL [16,33,64,77,127]. The recovery of a functional T_EM_ subset can be established within 1–2 months post-HCT [10,52], although their expansion can be dampened by posttransplant Cy [121]. T_EM_ numbers are higher from 2–6 months in recipients of haplo-HC grafts compared to both sibling and unrelated matched grafts, while recipients of unrelated matched grafts show higher total counts of T_EM_ than recipients of sibling matched grafts [10]. This is probably due to the rapid expansion of alloreactive T_EM_ or rapid proliferation of T_EM_ infused with the graft upon re-exposure to antigens [16]. On the other hand, as described for T_n_ cells, HLA mismatches in haplo-HCT might also impact T_EM_ reconstitution due to donor alloreactive NK cell cytolytic activity against DCs and T_EM_ cells [119,120]. Additional infusions of lymphocytes of the same donor (DLI) selected to deliver the graft may be used to increase the anti-leukemic effect and strengthen the protection against infections, as most cells possess a memory phenotype. However, DLI treatments come with the costs of increased GvHD toxicity, again exposing the delicate balance between GvL and GvHD responses.

The T_SCM_ subset is highly enriched following HCT, despite differences in conditioning regimen, cell source and GvHD prophylaxis [35,117,121]. However, T_SCM_ reconstitution seems to be influenced by the occurrence of GvHD [64]. T_SCM_ cell counts are significantly higher following CBT compared to BMT and PBT [35], which might be related to the higher number of naive T cells in CB grafts [29]. T_SCM_ cells are mainly derived from differentiation of non-alloreactive naive T cells infused with the graft due to a lymphopenic environment [117,121]. In addition, administration of posttransplant Cy contributes to the generation of T_SCM_ cells from naive precursors [117,121]. Differentiation of non-alloreactive naive T cells into T_SCM_ cells might play a crucial role in HCT outcome, as increased naive T cell counts are positively correlated with improved overall survival [2,15]. However, donor-derived alloreactive naive T cells might be able to differentiate into alloreactive T_SCM_ cells, contributing to GvHD [64]. As a high fraction of naive T cells differentiates into T_SCM_ cells after HCT, it might be valuable to include T_SCM_ cell markers in future studies to quantify T_SCM_ contribution.

Although our current knowledge on memory T cell reconstitution after HCT is limited, T_EM_ cells appear to reconstitute rapidly following HCT and may provide early immunological protection. Nevertheless, T_SCM_ cell reconstitution has been suggested to be most important, as this subset is able to differentiate into all memory and effector T cell subsets. More studies focusing on long-term reconstitution in large cohorts following both T-replete and T-depleted grafts are necessary to gain more insight into the reconstitution of this important T cell subset.

## 10. Concluding Remarks and Future Perspectives

Reconstitution of the T cell subsets is influenced by multiple transplantation- and patient-related factors and is highly correlated with HCT outcome (Figure 2). High numbers of Tregs [18,60,61], MAIT cells [74,94,95] and iNKT cells [102,103,104] correlate with protection from GvHD, while high Th and CD8^+^ T cell counts may be positively correlated with GvHD [2,16,45,73]. On the other hand, high numbers of donor-derived alloreactive Th and CD8^+^ T cells are associated with relapse-free survival [3,10]. Importantly, γδ T cells are not alloreactive and are therefore not associated with GvHD [34]. Moreover, γδ T cells are correlated with a lower incidence of opportunistic infections and a higher GvL effect [83,84]. All these different T cell subsets are encompassed within T_N_, T_EM_, T_CM_ and T_SCM_ cells. High concentrations of alloreactive T_n_ cells [25,45,64,116] and T_EM_ cells [16,35,127] infused with the graft exhibit strong anti-infectious and antileukemia effects but are also correlated with GvHD. T_N_ and T_SCM_ cells have been suggested to be most important for HCT outcome, as these subsets are able to differentiate into all memory and effector T cell subsets [2,15,35,45]. Nevertheless, adequate reconstitution of all distinct T cell subsets following allo-HCT is associated with increased overall survival [2,15]. Together, these data suggest that overall survival and event-free survival not only require a fast immune recovery of the immune cells but also that immune recovery needs to be diverse (e.g., TCR diversity) and balanced to achieve homeostasis and prevent immune dysregulation.

A major impact on T cell reconstitution is the use of immunosuppressive therapies to prevent or treat GvHD [7,8]. This results in loss of TCR repertoire diversity by severely depleting the donor T cells [41,132] and decreasing the thymus-dependent development of phenotypically naive T cells [8,20,111]. Although patients can recover approximately the same TCR diversity as healthy individuals [24,133], long-term survival might be influenced by the effect on both hematopoietic stem and progenitor cells. The remaining cells need to divide more frequently to replenish the hematological niche, leading to increased mutation accumulation, which could subsequently promote the development of secondary malignancies. This chance is probably increased after BMT and PBT, as the majority of mutations in hematopoietic stem cells are acquired after birth and accumulate gradually with age [134]. Furthermore, the limited renewing capacity of adult stem and progenitor cells might result in exhaustion of these cells and eventually result in a slower reconstitution [133].

Adjusting the GvHD prophylaxis or a better prediction and more selective treatment for patients at risk of developing GvHD determined by monitoring of (intentionally increased) the reconstitution of the distinct T cell subsets might improve the allo-HCT outcome. The total number of Treg, MAIT, iNKT and T_n_ cells in the grafts and/or early after transplantation may be used as predictive values for later development of GvHD. In addition, predicting MAIT cell reconstitution by monitoring B cells, diversity of gut microbiota and the amount of *ribA* and *ribB* genes in the microbiomes within the first months after allo-HCT might also serve as predictor of GvHD risk [92]. Novel therapeutic options to prevent GvHD can be early infusion of donor Tregs [135], promote Treg expansion by Sirolimus treatment [66,136] or intentionally activating and expanding iNKT cells [105]. Recently, a Phase 2A clinical trial showed that RGI-2001 administration was associated with reduced GvHD risk by increasing iNKT-cell induced Treg expansion early after allo-HCT [136]. A Phase 2 clinical trial evaluating the safety and efficacy of repeat doses of RGI-2001 doses is currently ongoing (Identification No. NCT04014790). Furthermore, selectively depleting CD45RA^+^ cells instead of all CD3^+^ cells in grafts might serve as a novel method to abolish serotherapy, reduce cGvHD and preserve infection protection and memory T cell reconstitution [112,125,137]. Together, immunosuppressive therapies to treat or prevent GvHD might be adjusted based on whether the T cell subsets adequately reconstitute after allo-HCT, resulting in a better clinical outcome.

Adequate CD4^+^ T cell reconstitution is strongly associated with increased survival chances in patients with adenovirus reactivation. One might consider that the use of preemptive antiviral therapies can be delayed or started from a higher viral load in patients with sufficient CD4^+^ T cell recovery at the time of viral reactivation. Especially taking antiviral drug toxicities into account, since timely CD4^+^ T cell reconstitution prevents viral reactivations and is correlated with a lower VR-associated mortality [39]. Furthermore, early monitoring of CD4^+^ T cell recovery provides the opportunity to identify patients at risk of viral reactivations and therefore to preemptively intervene with antiviral therapies. The recovery of CD4^+^ T cells itself following HCT might be predicted by monitoring early innate immune recovery, which has been shown to be positively correlated with CD4^+^ T cell recovery [53].

The positive correlation between T_n_ cell counts and adequate T cell reconstitution together with the knowledge that a large part of the immunological memory is derived from T_n_ cells [117,129,130] highlights the importance of the naive T cell compartment. This suggests a better immunological memory and long-term protection following CBT, which is additionally supported by the significantly higher T_SCM_ cell numbers observed in patients receiving CB grafts [35]. Nevertheless, immunological memory might also be maintained by division of already existing memory T cells, suggesting an important contribution of donor memory T cells in the life-long protection against pathogens following BMT and PBT. Furthermore, donor memory T cells are beneficial in protection against opportunistic infections and exerting GvL effects after transplantation [33,77,127], highlighting the important role of memory T cells post-HCT. These results, however, suggest a more crucial role for T_n_ cells compared with memory T cells in T cell reconstitution and thereby HCT outcome. More detailed knowledge on the yet undefined relation between T_n_ cell heterogeneity and later development of complications and mortality after transplantation is therefore necessary to improve HCT outcome.

The T cell compartment consists of a large variety of subpopulations which are all associated with distinct effector functions; however, most observational clinical studies focusing on T cell reconstitution are not able to discriminate between these cell subsets or effector cell functionality. These subsets are associated with distinct adverse or beneficial events; nevertheless, more detailed immune monitoring, including standardization of T cell subset identification between centers, is necessary to gain more insight into the biological mechanisms underlying immune reconstitution of the T cell compartment after allo-HCT. In conclusion, more in-depth knowledge of T cell reconstitution following transplantation will contribute to more effective treatment interventions, eventually leading towards more individualized approaches for patients undergoing allo-HCT.

## Figures and Tables

**Figure 1 cancers-12-01974-f001:**
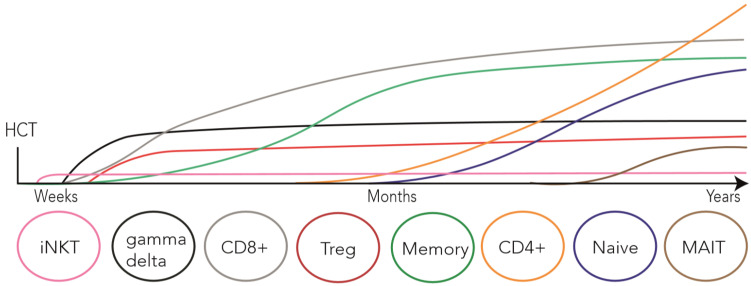
Schematic overview of the reconstitution of the distinct T cell subsets following allogeneic hematopoietic cell transplantation (HCT). iNKT, γδ T cells and Treg cells rapidly recover within weeks to normal levels after the time of Transplant. CD8^+^, CD4^+^ and memory T cell recovery can be as early as one to two months, and these subsets subsequently reconstitute within one to two years. Reconstitution of the naive T cell pool highly depends on thymopoiesis and can take years, starting around three months after transplantation. MAIT cell frequencies seem to remain extremely low within the first year and only reach normal levels after years following allogeneic HCT.

**Figure 2 cancers-12-01974-f002:**
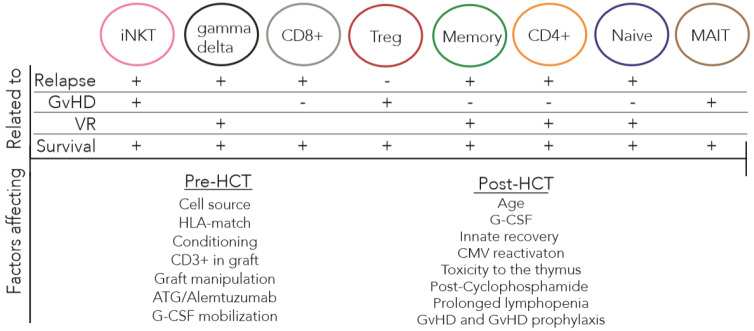
Overview of T cell subset related to allogeneic HCT outcomes and factors affecting T cell reconstitution before and after allo-HCT. Adequate reconstitution of the distinct T cell subsets is differently correlated to relapse, GvHD, VR and overall survival following allo-HCT. Adequate reconstitution of the T cell compartment is highly influenced by distinct factors before and after transplantation.

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
