# Peer review of "Reconstitution of T Cell Subsets Following Allogeneic Hematopoietic Cell Transplantation"

_cancers, 2020, doi:10.3390/cancers12071974_

Round 1

Reviewer 1 Report

The review proposed by Linde Dekker et al. is a great deal of interest for graft physicians and researchers in this field. It focuses on the post-HCT dynamics of different T cell reconstitution and their relations to outcome, considering different cell sources and immunosuppressive therapies.

While the topic is particularly rich and broad, the authors managed to address all aspects of the subject in maintaining a consistent and coherent whole. The review is well documented by 138 references.

Although authors included HLA-match in factors affecting T cell reconstitution, maybe HLA incompatibilities are not enough assessed in this review. Indeed, HLA mismatches impact on NK cell reconstitution. As NK cells are the first lymphocytes to shortly appear, their post-HSCT dynamic indirectly impacts on T cell reconstitution. This point should be added.

Moreover how ATG versus PTCY as GvHD prophylaxis impact on T cell reconstitution and their relations to clinical outcome should be more supplied.

In line with the fact that NK and T cell dynamics are nested, T cell subsets are also nested, notably Treg population regulates other lymphocyte populations. This point should be added.

There is a lot of uncertainly related to the numerous factors affecting T cell reconstitution. Although the authors tried to conclude on the main knowledge for each T cell subset, many questions raised opened. Thus, the concluding paragraph in each section is important and constitutes an added value.

I may propose a subtle arrangement. Indeed, Figure 2 should be more adapted in “Concluding remarks and future perspectives”, part in which the authors summarized in which clinical outcomes T cell subsets are engaged.

Remarks concerning editing:

Problems of editing for 2 references (7&84).

Modify “ab” by “ab” page 6 line 250

Author Response

We would like to thank the reviewer for the friendly and valuable comments. We really appreciate your time and effort. Here, we provide a point to point response to the comments. Please see the revised manuscript in the attachment. 

1)Although authors included HLA-match in factors affecting T cell reconstitution, maybe HLA incompatibilities are not enough assessed in this review. Indeed, HLA mismatches impact on NK cell reconstitution. As NK cells are the first lymphocytes to shortly appear, their post-HSCT dynamic indirectly impacts on T cell reconstitution. This point should be added.

Reply: This is a very interesting and valuable point. We added this point to

  • page 2, line 65
  • page 8, line 348-356
  • page 9, line 410-412

2) Moreover how ATG versus PTCY as GvHD prophylaxis impact on T cell reconstitution and their relations to clinical outcome should be more supplied.

Reply: We agree with this point, we added more on ATG versus PTCy to

  • page 3, line 119-121
  • page 4, line 154-157
  • page 5, line 212-214
  • page 9, line 405-406
  • page 9, line 421-422

3) In line with the fact that NK and T cell dynamics are nested, T cell subsets are also nested, notably Treg population regulates other lymphocyte populations. This point should be added.

Reply: A more detailed description of the Treg population in regulating other lymphocyte populations has been added on page 4, line 138-139.

4) I may propose a subtle arrangement. Indeed, Figure 2 should be more adapted in “Concluding remarks and future perspectives”, part in which the authors summarized in which clinical outcomes T cell subsets are engaged.

Reply: Thank you for this suggestion. We agree that this Figure is better suited in the “Concluding remarks and future perspectives” section. Figure 2 is therefore moved to page 10.  

Remarks concerning editing:

Problems of editing for 2 references (7&84).

Modify “ab” by “ab” page 6 line 250

Reply: Thank you for being attentive. Both remarks have been adjusted.

Reviewer 2 Report

Authors described about the review of reconstitution of T cell subsets following allogeneic hematopoietic cell transplantation. The manuscript is well written but some concern is left.

  1. Treg

Regulatory T cells (Tregs) have play a pivotal role in self-tolerance in mammalian. Authors should describe about general function of Tregs more in detail.

  1. CD8 T cell

Authors should describe about CMV-specific T cell and their immune response in hematopoietic stem cell transplantation.

  1. Naïve T cells

The naïve T cell (Tn) can contribute to GVHD and MEK/ERK signal are involved in the T cell differentiation. Authors should add the points.

Author Response

We would like to thank the reviewer for the friendly and valuable comments. We really appreciate your time and effort. Here, we will provide a point to point response to the comments. Please see the revised manuscript in the attachment. 

1) Regulatory T cells (Tregs) have play a pivotal role in self-tolerance in mammalian. Authors should describe about general function of Tregs more in detail.

Reply: A more detailed description has been added on the role of the Treg population in regulating other lymphocyte populations on page 4, line 138-139.

2) CD8 T cell. Authors should describe about CMV-specific T cell and their immune response in hematopoietic stem cell transplantation.  

Reply: Thank you for this valuable suggestion. Information on the relation between CMV-specific T cells and immune reconstitution has been added to page 5, line 193 and line 199-201.

3) Naive T cells. The naive T cell (Tn) can contribute to GVHD and MEK/ERK signal are involved in the T cell differentiation. Authors should add the points.

Reply: The statement that naive T cells can contribute to GvHD is mentioned on page 8, line 335-337, and page 10, line 442-444.

Although we agree that the involvement of MEK/ERK in T cell differentiation is valuable information, this signaling pathway is one of many pathways involved in T cell differentiation. We think that the description of a complete overview of signal transduction pathways is outside the scope of this review. We therefore suggest leaving the MEK/ERK signal, and other signaling pathways, outside this review.

Round 2

Reviewer 2 Report

No comment.